# Reducing mortality in acute kidney disease through comprehensive laboratory support: Findings from the ICLATA hybrid implementation study in Zambia

Sepiso K. Masenga[1,2,3]*, Luyando Mutelo[1,2], Cornelius Simutanda[1,2], Lukundo Siame[1,2], Gift C. Chama[1], Lweendo Muchaili[1,2], Bislom C. Mweene[1,2], Situmbeko Liweleya[1,2], Sydney Mulamfu[1,2], Benson M. Hamooya[1,2], Annet Kirabo[3,4,5]

1 HAND Research Group, School of Medicine and Health Sciences, Mulungushi University, Livingstone, Zambia, 2 Department of Cardiovascular Science and Metabolic Diseases, Livingstone Center for Prevention and Translational Science, Livingstone, Zambia, 3 Vanderbilt Institute for Global Health, Vanderbilt University Medical Center, Nashville, Tennessee, United States of America, 4 Department of Medicine, Vanderbilt University Medical Center, Nashville, Tennessee, United States of America, 5 Vanderbilt Center for Immunobiology, Vanderbilt Institute for Infection, Immunology and Inflammation, Vanderbilt University Medical Center, Nashville, Tennessee, United States of America

* sepisomasenga@gmail.com, sepisomasenga@lcpts.org

## Abstract

Acute kidney disease (AKD) contributes significantly to morbidity and mortality, particularly in low-resource settings where limited diagnostic capacity often leads to delayed recognition and suboptimal management. We aimed to evaluate the impact of implementing a comprehensive patient-specific clinical laboratory support on clinical outcomes in AKD and to identify factors associated with mortality. We conducted a mixed-method hybrid type 3 implementation study at Livingstone University Teaching Hospital in Zambia. The study compared a retrospective non-intervention cohort (NIC; n = 39) with a prospective intervention cohort (IC; n = 39) matched for age and sex. The intervention included providing full laboratory diagnostic support for AKD management and additional patient-specific tests. The primary outcome was death within 30 days after admission regardless of whether the patient was discharged before or after the thirty-day period elapsed. Data were analyzed using logistic regression and survival analysis. The median age of the NIC (43 years, IQR 31–53) was comparable to the IC (41 years, IQR 34–53), p = 0.996. Overall, 51.3% (n = 40/78) were males. Mortality rate was significantly lower in the intervention group, with deaths occurring in 5.1% of the IC compared to 28.2% in the NIC (p = 0.012). Logistic regression confirmed the intervention as a strong independent predictor of survival (adjusted odds ratio 0.07, p = 0.009). Diagnostic accuracy improved, with fewer cases of misdiagnosis or delayed diagnosis in the IC (7.7% vs 30.8%, p = 0.019). Re-hospitalization was significantly lower in the IC (38.6% vs 61.4%, p = 0.022). ESRD was more frequently recorded in the IC due to better diagnostics,

**Data availability statement:** All relevant data are within the paper and its Supporting Information files.

**Funding:** The author(s) declare that financial support was received for the research, authorship, and/or publication of this article. This work was supported by the International Federation of Clinical Chemistry and Laboratory Medicine (IFCC)'s Task Force on Outcome Studies in Laboratory Medicine (TF-OSLM) (to SKM), National Institute of Diabetes and Digestive and Kidney Diseases of the National Institutes of Health (2D43TW009744 to SKM; R21TW012635 to AK and SKM) and the American Heart Association (24IVPHA1297559 to AK and SKM). The funders had no role in study design, data collection and analysis, decision to publish, or preparation of the manuscript.

**Competing interests:** The authors have declared that no competing interests exist.

follow-up and survival. The median time-to-ESRD was substantially longer in the IC compared to NIC (140 vs 21 days, p < 0.0001). Implementation of a comprehensive patient-specific clinical laboratory support for AKD/CKD management significantly improved diagnostic precision and survival and reduced re-hospitalization. These findings highlight the value of strengthening laboratory diagnostic capacity to improve AKD outcomes in low-resource settings.

## 1 Introduction

Acute kidney injury (AKI), now conceptualized as part of a broader continuum of acute kidney diseases (AKD) is a major global health concern, affecting over 13 million people worldwide each year and contributing to nearly 1.7 million deaths [1–3]. About 80% of this burden occurs in low- and middle-income countries (LMICs), with sub-Saharan Africa (SSA) in particular shouldering a disproportionate share [4,5]. Clinical outcomes of AKI/AKD are especially poor in resource-limited settings. For example, a recent review reported pediatric AKI mortality of ~34% in Africa, more than double the ~13.8% mortality seen in children globally [6,7]. Zambia likewise faces a high acute kidney disease burden, as reported in one Zambian tertiary hospital, over half of ICU patients developed AKI (52.9% incidence) [1], reflecting the severe impact of acute kidney disorders in the region.

Multiple health-system challenges contribute to the poor outcomes of AKD in SSA. Patients often present late to care and advanced therapies such as dialysis are scarce, but equally important is the lack of timely diagnostic services. Many hospitals in low-resource settings cannot reliably perform or rapidly turnaround basic laboratory tests such as serum creatinine, resulting in delayed or missed diagnoses of AKD [7]. This suboptimal diagnostic capacity, alongside limited treatment options is a key factor underlying the high AKI/AKD mortality observed in SSA [7].

In recent years, nephrology guidelines have refined the clinical definitions of acute kidney syndromes to emphasize a continuum from injury to disease. Kidney Disease: Improving Global Outcomes (KDIGO) and the Acute Dialysis Quality Initiative (ADQI) have introduced the term "acute kidney disease" to encompass acute or subacute impairments in kidney function beyond the traditional scope of AKI [8]. KDIGO defines AKD as any abnormality of kidney function and/or structure of ≤3 months' duration that does not meet criteria for chronic kidney disease (CKD) [8]. By this definition, AKI (an acute decline usually within 7 days) is a subset of AKD, but importantly AKD also includes acute kidney dysfunction that does not strictly fulfill AKI criteria; such AKD without overt AKI is relatively common and is associated with higher risks of progression to end-stage kidney disease and mortality [8,9]. Meanwhile, ADQI consensus recommendations propose using a 7-day timeframe to distinguish AKI from AKD, whereby persisting renal dysfunction beyond 7 days after an AKI episode is classified as AKD to denote incomplete recovery in the subacute phase [8,9]. These evolving classifications underscore that AKD represents a broader clinical continuum bridging acute injury and chronic kidney disease, highlighting the need to recognize and manage kidney dysfunction even when it falls outside classic AKI criteria.

Early detection and management of AKI/AKD are critical to improving patient outcomes, as prompt interventions can prevent progression to more severe or irreversible disease [10,11]. Key to early diagnosis is the availability of reliable laboratory testing. Clinicians require timely measurements of renal function to confirm AKD, assess its severity, and monitor the patient's course [12]. Conversely, in settings where basic tests are unavailable or delayed, opportunities for early intervention may be missed, leading to more advanced disease presentation and worse prognoses. Indeed, international initiatives such as the International Society of Nephrology's "0by25" program stress that AKI is often preventable and treatable if identified early, calling for improved diagnostic capacity in disadvantaged populations [13].

In light of these challenges, there is a compelling rationale to strengthen laboratory diagnostic capacity for AKD in resource-limited health systems. We hypothesized that providing tailored clinical lab test support for each AKD patient would enable earlier diagnosis, better management, and improved outcomes. However, to date there is little published evidence from SSA regarding the impact of such diagnostic interventions. To address this gap, the ICLATA study was undertaken to evaluate the effect of introducing a comprehensive patient-specific clinical laboratory support on clinical outcomes in AKD and to identify factors associated with mortality. Specifically, we aimed to assess whether the availability of a full kidney test profile at the point of care would lead to improved AKD outcomes such as reduced mortality, enhanced recovery of renal function, and less progression to chronic kidney disease compared to the standard limited test availability.

## 2 Method

### 2.1 Ethics statement

Ethical approval was obtained from Mulungushi University School of Medicine and Health Sciences Research Ethics Committee (MUHSREC) (Assurance No. FWA0002888 IRB00012281 of IORG0010344 on the 11th of August 2023 and the National Health Research Authority (NHRA) on the 23rd of August 2023. Permission to conduct research was given by the LUTH's administration. Written consent was waived by the ethics committee for the retrospective NIC where secondary data was used. For the IC, we obtained written consent from the participants before enrolling them in the study. No personal identifying data were obtained.

### 2.2 Study design and setting

We conducted a mixed model hybrid type 3 implementation study at Livingstone University Teaching hospital (LUTH) in Zambia. The study compared two cohorts of adults (≥18 years) with AKD. The Non-Intervention Cohort (NIC) was a retrospective group (n = 39) identified from 1,136 patient records in 2021–2022; only 3.4% (39/1136) met inclusion criteria of AKD at admission with complete data on required outcomes and labs. The Intervention Cohort (IC) was a prospective group (n = 39) enrolled from September 4, 2023 to March 7, 2024. We matched the intervention cohort to the non-intervention cohort on a one-to-one basis by sex and by age (±5 years). All participants were adults (≥18 years old).

### 2.3 Intervention

In the IC, we implemented the intervention of providing comprehensive laboratory testing support for AKD diagnosis and management. This complete laboratory profile included kidney function tests (serum creatinine, urea, uric acid, electrolytes), liver function tests (ALT, AST, proteins, bilirubin, alkaline phosphatase, GGT), urine tests (e.g., microalbumin), full blood counts, HbA1c, infectious disease screens (HIV, hepatitis B/C, syphilis), cardiac biomarkers (CK-MB, troponins), fasting insulin, lipid profile, and any other tests needed for individual patient care. The IC had continuous laboratory support throughout hospitalization, not just a one-time test, enabling ongoing assessment. Testing was tailored and iterative rather than one-size-fits-all. While not exhaustive by high-income standards, this panel represents the minimum diagnostic support required to optimally manage AKD in our low-resource setting. By contrast, the NIC patients received the standard care available at the time, which lacked many reagents and tests and laboratory support was inconsistent or insufficient for AKD management. In the IC, laboratory tests were performed and resulted without delay, typically within a few hours

on the same day of ordering, in contrast to the NIC era where tests (if available at all) often had significant delays (24–72 hours, sometimes up to 7 days) or were not done until a crisis arose. For the IC patients, our study team coordinated closely with the laboratory to ensure expedited sample transport and processing. Blood samples from IC patients were prioritized and run immediately on the analyzers, and results were relayed to the treating team as soon as they were ready. In the NIC, by contrast, clinicians frequently had to wait for reagents to be procured or batch testing to be done, leading to diagnostic and treatment delays.

## 2.4  Follow up period

The primary outcome was 30-day mortality, meaning we monitored survival status for 30 days after admission for each patient regardless of whether they had been discharged earlier. Additionally, for secondary outcomes like progression to CKD or ESRD, as well as rehospitalization, we collected data for up to 3 months after the initial AKD diagnosis, in line with the 90-day definition of AKD. In the prospective IC, patients were followed through outpatient visits or phone calls up to 90 days and in some cases beyond, up to ~6 months for those who consented to extended follow-up to document longer-term outcomes. The IC participants were followed until July 31, 2024, with a median follow-up of approximately 3 months, whereas the retrospective NIC data were limited to what was documented in their records, typically up to discharge and any return visits within 3 months. Thus, the follow-up for mortality was 30 days for all, and for other outcomes up to 90 days.

In our setting (Livingstone, Zambia), chronic dialysis resources were available but extremely limited. Not every patient who progresses to ESRD can receive long-term dialysis often due to resource and access constraints and there are no kidney transplant services.

## 2.5  Eligibility of participants and study variables

All participants met criteria for AKD. AKD was defined as acute or subacute abnormalities of kidney function/structure beyond 7 days and up to 3 months, including but not limited to those fulfilling AKI criteria [14,15]. We used serum creatinine changes and clinical assessment to diagnose AKD, in line with KDIGO AKI guidelines and emerging AKD definitions [14,15]. NIC patients were diagnosed clinically and with whatever few tests were on hand, whereas IC patients had the benefit of complete diagnostic workups. Patients with missing key data such as age, sex, AKD diagnosis, hypertension status, recovery status or outcomes were excluded. The primary outcome was death. We defined AKI per KDIGO criteria (serum creatinine rise ≥0.3 mg/dL within 48h or ≥1.5 × baseline within 7 days, or urine output <0.5 mL/kg/h for 6h [12]). We defined acute kidney disease (AKD) as an abnormality in kidney function and/or structure persisting >7 days but ≤3 months [14,15]. Chronic kidney disease (CKD) was defined by an estimated GFR < 60 mL/min/1.73m$^2$ or albuminuria ≥30 mg/24h, or other markers of kidney damage for >3 months, and ESRD as kidney failure with GFR < 15 mL/min/1.73m$^2$ requiring renal replacement therapy [16]. The presence of underlying CKD was determined by a documented clinical history of CKD, evidence of chronically impaired kidney function from previous laboratory records, or, for newly diagnosed patients, by the persistence of kidney damage markers beyond the 3-month AKD timeframe during the study's follow-up period. Biochemical tests (such as creatinine, urea, electrolytes, liver enzymes, etc.) were performed on an automated Pentra C400 Clinical Chemistry Analyzer in our hospital laboratory. For electrolytes, we used a HumaLyte Plus 5 - Ion-selective electrolyte analyzer. Hematological tests (complete blood counts) were run on an automated hematology analyzer, Sysmex XT-2000i. For certain specialized tests like HIV, hepatitis serologies, and syphilis, we specify that rapid diagnostic test kits (lateral flow assays) or ELISA were used as per the hospital's standard procedure.

Independent variables included socio-demographics (age, sex, marital status, employment), clinical factors (hypertension, HIV status, suspected AKD cause, signs and symptoms, presence of renal infection, inpatient vs outpatient status, misdiagnosis or delayed diagnosis, hospital re-admission), laboratory results, and medication costs. Data for the NIC were

abstracted from paper patient files, while IC data came from patient files supplemented by the laboratory information system. Data collection spanned from 1st September 2023–31st July 2024 for both cohorts.

## 2.6 Bias

To minimize bias, we addressed the significant missing data in NIC records by excluding files lacking critical outcomes or lab results particularly missing baseline kidney function tests that would preclude comparisons.

## 2.7 Sample size assumptions and estimation

We performed a census sampling of all AKD patients available in the NIC (yielding 39 patients) and intentionally matched the IC sample size (39 patients) with similar age and sex distribution (1:1 ratio) for comparability. Post-hoc, we conducted a power analysis using the observed outcome rates in NIC, an uncorrected chi-square test for two independent proportions indicated ~89.7% power ($\alpha = 0.05$) to detect the observed difference suggesting our sample was sufficient to support the main findings.

## 2.8 Statistical methods

We used statCrunch (https://www.statcrunch.com), a web-based statistical software for analysis. Continuous variables were summarized with medians and interquartile ranges (IQR) and compared using Mann-Whitney U tests since data were non-normal. Categorical variables were summarized as frequencies and percentages. Group differences were assessed with Chi-square or Fisher's exact tests as appropriate. Univariable and multivariable logistic regression models were constructed to identify factors associated with the intervention (IC vs NIC membership) and with mortality as an outcome of interest. Variables significant at $p < 0.05$ and if clinically significant in univariable analysis were considered for entry into multivariable models. For the mortality analysis, we report odds ratios (OR) and adjusted OR (AOR) with 95% confidence intervals. We also performed a Kaplan-Meier survival analysis comparing time-to-death between cohorts, using the log-rank test to compute hazard ratios (HR). A two-tailed $p < 0.05$ was considered statistically significant for all tests.

This report adheres to the STROBE (strengthening the reporting of observational studies in epidemiology) guidelines for cohort studies, S1 STROBE checklist in S1 Checklist.

# 3 Results

## 3.1 Study population

Out of 1,136 patient records screened in the retrospective period, only 39 patients (3.4%) met eligibility for AKD. These 39 NIC patients were matched to 39 IC patients prospectively enrolled, yielding a total sample of 78 participants, Fig 1. The cohorts were well matched in core demographics (Table 1). The overall median age was 41 years (IQR 32–53). NIC median age was 43 (31–53) vs IC 41 (34–53), p = 0.996, indicating no age difference. There were slightly more males (51.3%) than females overall, with similar sex distribution in NIC (52.5% male) and IC (47.5% male), p = 0.651. Two-thirds of patients (66.7%) had a history of hypertension, with no significant difference between NIC and IC (NIC 46.2% vs IC 53.8% hypertensive, p = 0.336). HIV status was known for 70 patients; 28.6% were HIV-positive, equally distributed between NIC and IC (each 50% of persons living with HIV, p = 0.542).

## 3.2 Clinical presentation and comorbidities

There were notable differences in underlying causes of AKD between cohorts (Table 1). Overall, hypertension (often malignant or acute hypertensive emergency) was the leading suspected cause of kidney injury (34.6% of participants). This was far more frequent in the IC (77.8%) than in the NIC (22.2%), p < 0.0001, suggesting better recognition of

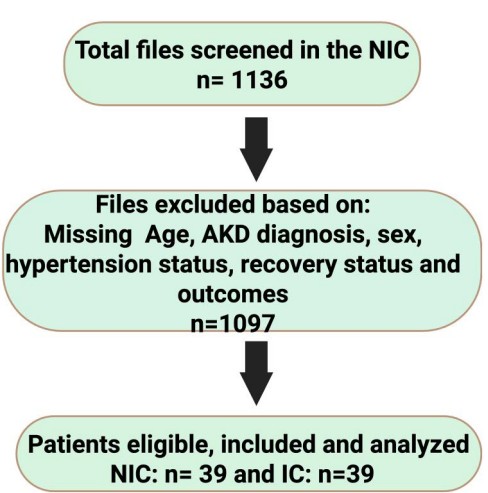

**Fig 1. Flowchart of eligibility and selection of participants.** This figure illustrates the selection of participants. Initially 1,136 patient records were reviewed for the NIC; only 39 met inclusion criteria (AKD diagnosis and complete data). These 39 were matched to 39 newly enrolled AKD patients in the IC. Thus, both cohorts had equal numbers (n = 39) for comparison. AKD, acute kidney injury; IC, interventional cohort; NIC, non-interventional cohort.

hypertension-related AKD when lab support was available. Conversely, acute pyelonephritis was documented in 11.5% of participants, all in the NIC and none in the IC (NIC 9 participants vs IC 0, p < 0.0001), implying that in the IC, what would have been diagnosed as pyelonephritis might have been managed differently or prevented, or simply that those infections did not occur in the smaller IC sample. Other etiologies (kidney stones 11.5% overall; severe anemia 9%; sepsis, bladder cancer, systemic lupus erythematosus (SLE), or tuberculosis each ~5%) were observed in both cohorts in a few cases (numbers too small for individual significance testing). Overall, having a known identifiable cause of the kidney insult was significantly more common in the IC (60.7% of patients) than in the NIC (39.3%, p = 0.002). This highlights that the intervention's comprehensive testing enabled clinicians to pinpoint an etiology for AKD more often, for example, confirming autoimmune disease, infections, or other organ dysfunction, whereas in the NIC many cases remained of unknown cause.

Similarly, newly diagnosed or pre-existing chronic kidney disease (CKD) was prevalent (70.5% of all patients had evidence of CKD in addition to AKD). This was significantly higher in the IC as 60% of those with CKD were in the IC vs 40% in NIC (p = 0.006). The greater detection of CKD in IC likely reflects the availability of tests (e.g., for baseline renal function, proteinuria) that identified chronic disease which may have been missed in NIC. Exposure to nephrotoxic drugs was common (38.4% overall) with no difference between cohorts (NIC 41.0% vs IC 35.9%, p = 0.641). Notably, misdiagnosis or delayed diagnosis of the kidney condition was significantly more frequent in NIC: 19.2% of all patients experienced an initial misdiagnosis or delay, and of those, 80% were in NIC (12 participants) vs only 20% in IC (3 participants), p = 0.019. In other words, nearly all misdiagnosed cases occurred in the absence of timely lab support. This underscores the impact of the intervention on diagnostic accuracy. We also observed a difference in the setting of presentation: the IC had a higher proportion of outpatients (63.6%) at the time of AKD diagnosis, whereas the NIC had more inpatients (60.0%) (p = 0.039). This may suggest that with better lab screening, more AKD cases were identified and managed without requiring hospital admission (or it could reflect differences in referral patterns between the time periods).

Baseline laboratory values for key electrolytes were only reliably available for the IC (since NIC often lacked these tests). In the IC, median baseline serum sodium was 140.0 mmol/L (IQR 137.0–145.0), potassium 4.00 mmol/L (3.50–5.45), and chloride 97.0 mmol/L (94.0–101.0). These data were not available for NIC.

**Table 1. Baseline sociodemographic and clinical characteristics of the non-intervention and intervention cohorts.**

| Variables | Frequency (%) | Non-intervention cohort (NIC), n = 39 | Intervention cohort (IC), n = 39 | P-Value |
|---|---|---|---|---|
| **Age,** *years, or Median (IQR)* | 41 (32 - 53) | 43 (31 - 53) | 41 (34 - 53) | 0.996 |
| **Sex** | | | | |
| *Male* | 40 (51.3) | 21 (52.5) | 19 (47.5) | 0.651 |
| *Female* | 38 (48.7) | 18 (47.4) | 20 (52.6) | |
| **Hypertension** | | | | |
| *No* | 26 (33.3) | 15 (57.7) | 11 (42.3) | 0.336 |
| *Yes* | 52 (66.7) | 24 (46.2) | 28 (53.8) | |
| **People with HIV** | | | | |
| *No* | 50 (71.4) | 21 (42) | 29 (58) | 0.542 |
| *Yes* | 20 (28.6) | 10 (50) | 10 (50) | |
| **Suspected Underlying cause of AKD** | | | | |
| *Hypertension* | 27 (34.6) | 6 (22.2) | 21 (77.8) | **<0.0001** |
| *Pyelonephritis* | 9 (11.5) | 9 (100.0) | 0 (0.0) | |
| *Kidney stones* | 9 (11.5) | 3 (33.3) | 6 (66.7) | |
| *Severe anemia* | 7 (9.0) | 2 (28.6) | 5 (71.4) | |
| *Other** | 4 (5.2) | 2 (50.0) | 2 (50.0) | |
| **Known underlying cause** | | | | |
| *No* | 22 (28.2) | 17 (77.3) | 5 (22.7) | **0.002** |
| *Yes* | 56 (71.8) | 22 (39.3) | 34 (60.7) | |
| **CKD** | | | | |
| *No* | 23 (29.5) | 17 (73.9) | 6 (26.1) | **0.006** |
| *Yes* | 55 (70.5) | 22 (40.0) | 33 (60.0) | |
| **Nephrotoxic drugs[c]** | 30 (38.4) | 16 (41.0) | 14 (35.9) | 0.641 |
| **Misdiagnosis/delayed diagnosis** | | | | |
| *No* | 63 (80.8) | 27 (42.9) | 36 (57.1) | **0.019[F]** |
| *Yes* | 15 (19.2) | 12 (80.0) | 3 (20.0) | |
| **Type of Patient** | | | | |
| *Out-patient* | 33 (42.3) | 12 (36.4) | 21 (63.6) | **0.039** |
| *In-patient* | 45 (57.7) | 27 (60.0) | 18 (40.0) | |
| **Baseline serum Na$^+$** | | – | 140.0 (137.0-145.0) | |
| **Baseline serum K$^+$** | | – | 4.00 (3.50-5.45) | |
| **Baseline serum CL** | | – | 97.0 (94.0-101.0) | |
| **Needed dialysis** | | | | |
| *No* | 25 (32.1) | 16 (64.0) | 9 (36.0) | 0.089 |
| *Yes* | 53 (67.9) | 23 (43.4) | 30 (56.6) | |
| **Re-hospitalized** | | | | |
| *No* | 34 (43.6) | 12 (35.3) | 22 (64.7) | **0.022** |
| *Yes* | 44 (56.4) | 27 (61.4) | 17 (38.6) | |
| **Number of days hospitalized** *n = 44* | 8 (5-16) | 7 (2-15) | 8 (7-23) | 0.060 |
| **Recovery time in days** | 14 (10 – 20) | 14 (10 – 20) | 16 (10 – 25) | 0.826 |
| **ESRD** | 46 (59.0) | 15 (32.6) | 31 (67.4) | **0.0002** |
| **Days to ESRD after AKD or CKD diagnosis,** n = 28 | 134 (83.5 – 188) | 21 (20 – 21) | 140 (100 – 189) | **<0.0001** |
| **Died** | | | | |
| *No* | 65 (83.3) | 28 (43.08) | 37 (56.9) | **0.012[F]** |
| *Yes* | 13 (16.7) | 11 (84.6) | 2 (15.4) | |

*Sepsis, bladder CA, Systemic lupus erythematosus, TB;**hypertensive emergency n = 3/8; F = fisher's exact test, [c]column percentage; ESRD, end-stage renal disease; CKD, chronic kidney disease; AKD, acute kidney disease.

### 3.3 Clinical outcomes

A majority of patients (67.9%) required dialysis at some point during management of AKD (determined by standard clinical indications such as uremic symptoms, refractory electrolyte imbalances, or fluid overload). The proportion of patients meeting dialysis criteria was high in both cohorts and not significantly different (NIC 43.4% of those needing dialysis vs IC 56.6%, p = 0.089), indicating that the need for renal replacement therapy was common regardless of intervention. However, the re-hospitalization rate (hospital readmission after initial discharge) differed: 56.4% of patients were readmitted at least once, and this was significantly less frequent in the IC (38.6%) compared to NIC (61.4%), p = 0.022. Thus, patients with lab-supported management were less likely to require coming back to the hospital, suggesting better stabilization or outpatient management. Among those who were re-hospitalized (n = 44), the median total length of hospital stay was 8 days (IQR 5–16). There was a trend toward longer stays in IC re-admissions (median 8 days, IQR 7–23) than NIC (7 days, IQR 2–15), but this difference did not reach statistical significance (p = 0.060). The median time to clinical recovery (from diagnosis to resolution of AKD or discharge) was similar: 14 days overall (IQR 10–20), with no difference between NIC (median 14 days) and IC (16 days), p = 0.826.

Progression to end-stage renal disease (ESRD) was a major adverse outcome observed in the study. Overall, 46 patients (59.0%) progressed to ESRD (meaning permanent kidney failure requiring long-term dialysis or transplant). Interestingly, a significantly higher proportion of these were from the IC (67.4%) compared to NIC (32.6%, p = 0.0002). On face value this suggests more IC patients ended up in ESRD, but this is likely due to greater survival and better follow-up in the IC, allowing time to reach the ESRD endpoint whereas some NIC patients may have died before ESRD could manifest or were lost to follow-up without documentation of progression. Supporting this interpretation, the median time from AKD/CKD diagnosis to ESRD among those who did progress (n = 28 with documented dates) was dramatically longer in the IC: 140 days (IQR 100–189) versus just 21 days in the NIC (IQR 20–21 days), p < 0.0001. In other words, in NIC participants that progressed to ESRD, it happened very quickly (within ~3 weeks of AKD diagnosis), likely because these were fulminant cases or acute-on-chronic scenarios identified late. In contrast, IC patients who progressed did so over several months, implying a slower progression possibly due to better-managed care or earlier stage detection.

### 3.4 Mortality

Overall mortality during the study period was 16.7% (13 out of 78 patients died). Strikingly, mortality was much lower in the IC. Among the 13 deaths, 11 (84.6%) occurred in NIC patients and only 2 (15.4%) in IC (p = 0.012). This translates to a case fatality of 28.2% in NIC vs 5.1% in IC. The dramatic reduction in deaths with the intervention is a key finding.

### 3.5 Laboratory characteristics

Baseline laboratory comparisons between NIC and IC are shown in Fig 2. Overall, the median initial serum creatinine was very high at 460.1 µmol/L (IQR 153.7–841.4), reflecting severe kidney impairment in this AKD population. The NIC had a median creatinine of 574.1 µmol/L (IQR 119.1–1163.3) versus 450.9 µmol/L (209.4–656.1) in the IC, but this difference was not statistically significant (p = 0.325). Baseline plasma urea was similarly elevated: 12.30 mmol/L overall (IQR 6.44–18.95), with NIC 10.19 (3.89–18.37) vs IC 12.20 (8.40–20.40), p = 0.367. The median estimated GFR (eGFR) at baseline was 8.76 mL/min/1.73m$^2$ overall (reflecting that many patients already had severely reduced kidney function). Median eGFR was 8.76 in NIC vs 12.8 in IC; this slight difference (NIC 8.8 vs IC 12.8) was not significant (p = 0.940). Liver enzymes: median alanine aminotransferase (ALT) was 20.6 U/L overall (IQR 10.3–33.0); by cohort, NIC 18.5 (10.3–20.6) vs IC 23.0 (19.0–34.8), p = 0.114. Aspartate aminotransferase (AST) median was 32.4 U/L overall (26.1–60.3); NIC 30.2 (26.1–34.7) vs IC 35.5 (27.5–192.5), p = 0.194. These liver function values suggest no significant hepatic injury differences; medians were within normal ranges for ALT/AST in both groups.

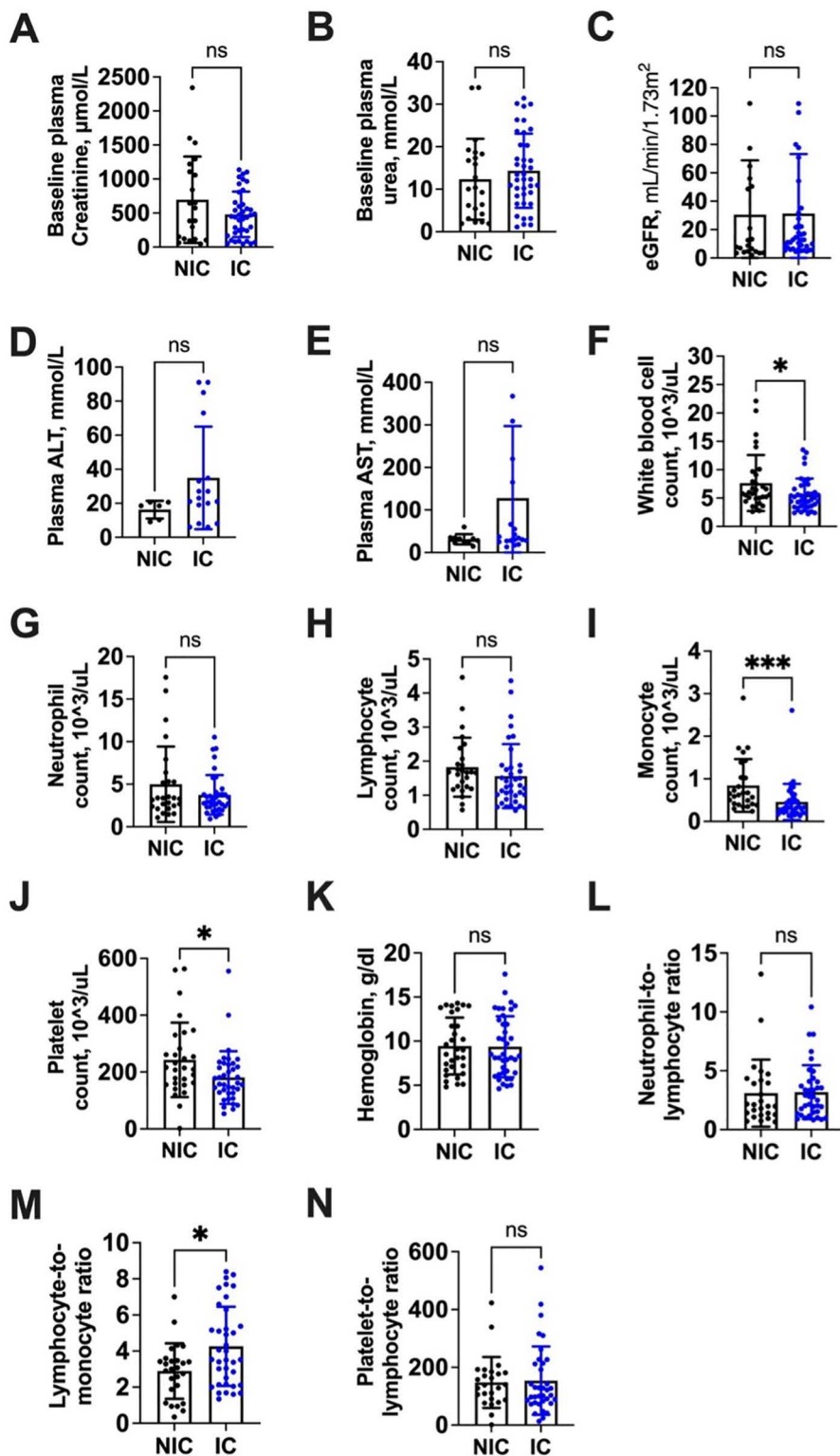

**Fig 2. Comparison of laboratory characteristics between the non-intervention and intervention cohorts.** Panel A: Baseline median creatinine (μmol/L) – NIC vs IC; B: Baseline plasma urea (mmol/L); C: eGFR (mL/min/1.73m²); D: ALT (U/L); E: AST (U/L); F: White blood cell count (×10^3/μL); G: Neutrophil count; H: Lymphocyte count; I: Monocyte count; J: Platelet count; K: Hemoglobin (g/dL); L: Neutrophil-to-lymphocyte ratio; M:

Lymphocyte-to-monocyte ratio; N: Platelet-to-lymphocyte ratio. Error bars or IQR ranges shown. Notable differences: NIC had significantly higher WBC (F), monocytes (I), platelets (J); IC had higher LMR (M). Other measures did not differ significantly between cohorts.).

For blood counts, some significant differences emerged. White blood cell (WBC) count was higher in NIC: median 5.8 × 10^3/μL (IQR 4.9–9.1) in NIC vs 4.9 (3.6–7.2) in IC, p = 0.031. This may indicate more inflammatory or infectious processes in NIC patients. Neutrophil count medians were 3.33 × 10^3/μL (NIC) vs 2.94 (IC), p = 0.759 (no difference). Lymphocyte count: 1.68 × 10^3/μL (NIC) vs 1.29 (IC), p = 0.253 (no significant difference). Monocyte count was notably higher in NIC: 0.64 × 10^3/μL vs 0.33 in IC, p = 0.005. Platelet count was also significantly higher in NIC: median 214 × 10^3/μL vs 159 in IC, p = 0.029. The NIC's higher WBC, monocyte, and platelet counts may reflect an acute inflammatory response (e.g., infection or stress response) more often in NIC patients. Hemoglobin levels were low in both groups (median 8.5 g/dL overall, indicating anemia was common). There was no difference: NIC 9.0 vs IC 8.3 g/dL, p = 0.920. Neutrophil-to-lymphocyte ratio (NLR) was high overall (median ~2.16) with no difference (NIC 2.05 vs IC 2.78, p = 0.893). Lymphocyte-to-monocyte ratio (LMR) was significantly higher in IC: median 3.53 vs NIC 3.04, p = 0.003. A higher LMR in IC aligns with lower monocytes and slightly lower lymphocytes, suggesting less systemic inflammation. Platelet-to-lymphocyte ratio (PLR) showed no difference (NIC 139 vs IC 113, p = 0.814). In summary, NIC patients had higher WBCs, monocytes, and platelets than IC patients at presentation, while IC patients had a higher LMR. These lab disparities suggest that the NIC might have included more patients with infection or inflammation (consistent with the fact that pyelonephritis cases were only in NIC), whereas the IC patients, many being outpatients, might have had less inflammatory stress.

## 3.6 Factors associated with mortality

We examined predictors of death in the combined cohort using logistic regression (Table 2). In univariate analysis, several variables showed potential association with mortality. Intervention (IC vs NIC) was the most striking factor such that being in the intervention cohort was associated with sharply reduced odds of dying. Unadjusted, the odds of death for IC patients were about 87% lower than for NIC patients (OR 0.13, 95% CI 0.02–0.67, p = 0.014). After adjusting for other variables in a multivariable model, the protective effect of the intervention remained highly significant and even stronger (AOR 0.07, 95% CI 0.01–0.53, p = 0.009). This suggests that even when controlling for baseline differences, the comprehensive lab support independently reduced mortality by roughly 93%. Among laboratory values, higher hemoglobin was associated with lower mortality risk in univariate analysis (OR 0.78 per 1 g/dL increase, 95% CI 0.61–0.99, p = 0.046), implying anemia was linked to higher death risk. In the adjusted model, hemoglobin's effect was similar in magnitude (AOR 0.72) but not statistically significant (p = 0.075), likely due to limited sample size. Initial serum creatinine level showed a very small univariate association with mortality (OR ~1.001 per μmol, p = 0.013), but this did not remain significant after adjustment (AOR ~1.00, p = 0.286) indicating that baseline creatinine by itself did not independently predict death once intervention and other factors were accounted for. Monocyte count was another factor: higher monocyte count was associated with higher odds of death in univariate analysis (OR 2.85 per unit, p = 0.049), suggesting that patients with an ongoing

**Table 2. Factors associated with mortality in univariable and multivariable logistic regression analyses.**

| Variables | OR (95%CI) | P-Value | AOR (95%CI) | P-Value |
|---|---|---|---|---|
| First Serum creatinine | 1.00 (1.00 − 1.00) | **0.013** | 1.00 (0.99 − 1.00) | 0.286 |
| Monocyte count | 2.85 (1.00 − 8.11) | **0.049** | 2.27 (0.59 − 8.79) | 0.231 |
| Hemoglobin | 0.78 (0.61 − 0.99) | 0.046 | 0.72 (0.51 − 1.03) | 0.075 |
| Intervention | | | | |
| *no* | 1 | | 1 | |
| *Yes* | 0.13 (0.02 − 0.67) | **0.014** | 0.07 (0.01 − 0.53) | **0.009** |

inflammatory response or sepsis (who tend to have elevated monocytes) had worse outcomes. However, monocyte count's effect was not significant in the adjusted model (AOR 2.27, p = 0.231), likely due to collinearity with the intervention (recall NIC had higher monocytes and also higher mortality). No other variables showed significant associations in the final model aside from the intervention.

### 3.7 Survival analysis

To further illustrate the survival benefit of the intervention, we performed a Kaplan-Meier survival analysis (Fig 3). The hazard ratio for mortality in NIC vs IC was 8.32 (95% CI 2.74–25.26), meaning patients in the NIC had over 8-fold higher risk of dying at any given time than those in the IC, p = 0.001 by log-rank test. The mean survival time was 16 days for NIC patients versus 26 days for IC patients (p = 0.0014). The survival curve diverged early, with most NIC deaths occurring in the first few weeks of follow-up, whereas IC mortality was not only lower in frequency but also tended to occur later. This analysis reinforces that the intervention significantly prolonged survival in patients with AKD.

## 4  Discussion

In this hybrid implementation study, we found that providing a comprehensive patient-specific clinical laboratory support for AKD patients led to markedly improved clinical outcomes, most notably a dramatic reduction in mortality. To our knowledge, this is one of the first studies to directly demonstrate that enhancing diagnostic capabilities in AKD management can save lives in a resource-limited hospital setting. Our results align with the broader understanding that timely recognition and appropriate management of acute kidney conditions are critical to preventing deaths [17]. In the NIC (limited-lab) cohort, over one-quarter of patients died, consistent with reports of high AKI/AKD mortality in low-resource contexts [17]. By contrast, in the IC (lab-supported) cohort, mortality was only ~5%, a rate approaching outcomes in well-resourced settings for comparable acute kidney injury severity [17].

The primary objective of our study was to identify factors associated with mortality and to evaluate differences between the intervention and control cohorts. The intervention itself emerged as the most significant independent factor associated with survival. Even after adjusting for potential confounders, patients who received comprehensive lab support had a ~93% reduction in the odds of death (AOR 0.07) compared to those who did not. This suggests a causal relationship. Providing clinicians and patients with timely lab results for kidney function, electrolytes, underlying disease markers, and additional patient-specific tests enabled more effective and targeted treatment, thereby preventing many deaths. This finding is in line with the conceptual goals of global initiatives like the International Society of Nephrology's "0by25" program, which advocates that no one should die of untreated AKI/AKD and emphasizes the need for point-of-care diagnostic tools in low-resource settings [13,18]. Our study provides empirical evidence supporting that philosophy, when diagnostic tools (in our case, a broad lab panel) were made available at LUTH, patient survival improved substantially.

**Fig 3.  Kaplan-Meier survival curve comparing NIC vs IC.** The IC group shows significantly better survival; log-rank p = 0.0014. The large hazard ratio (>8) indicates substantially higher mortality hazard without the lab-support intervention.).

## 4.1 Benefits of the intervention and lessons for improvement

Beyond survival, the intervention conferred several other benefits. Diagnostic accuracy improved. Only 3 patients in the IC experienced a misdiagnosis or delayed diagnosis of their kidney condition, versus 12 in the NIC. This eight-fold reduction corresponds with the idea that AKI/AKD often "goes unrecognized" when laboratory confirmation is lacking [17]. In practical terms, NIC patients without labs might have been diagnosed late (or misdiagnosed as something else) until kidney failure was far advanced, whereas IC patients were promptly identified as AKD, allowing earlier intervention. Similarly, the underlying cause of AKD was identified in 61% of IC patients compared to just 39% in NIC, a significant difference. In the NIC, many cases were labeled with non-specific diagnoses or the cause remained unknown, likely due to limited diagnostic testing. In the IC, having a "complete laboratory profile support" meant clinicians could uncover hidden etiologies, for example, detecting autoimmune disease through available serologies, confirming infections, or recognizing hypertensive emergency through proper workups. Indeed, hypertension-related AKD was recognized four times more often in IC than NIC. This may be because, with full lab support, physicians could differentiate intrinsic hypertensive renal injury from other causes, or perform workups such as urine analysis for protein, fundoscopic exam documentation, etc. to confirm hypertensive nephropathy. The NIC, lacking these tools, might under-diagnose hypertension as a cause. Conversely, NIC had several cases of pyelonephritis listed as AKD cause, whereas IC had none possibly indicating that in IC, better prophylaxis or early treatment prevented such infections, or that some NIC cases attributed to pyelonephritis might have been misdiagnosed due to absent confirmatory tests (e.g., urine cultures, imaging were likely scarce).

Another clear benefit was seen in hospital readmissions. IC patients were significantly less likely to be re-hospitalized (39% vs 61%). This suggests that the initial management in IC was more effective at stabilizing patients and possibly arranging appropriate follow-up such as treating the correct diagnosis definitively or optimizing kidney function recovery with proper lab-guided therapy, thereby reducing the need for return admissions. In NIC, by contrast, incomplete initial treatment or unresolved issues owing to diagnostic limitations may have led to more complications or progression, necessitating readmission. A related point is that IC had more outpatients and NIC more inpatients at baseline. It is conceivable that with comprehensive labs, some AKD cases were caught early and managed without admission (e.g., adjusting medications, outpatient dialysis for stable patients, etc.), whereas in the NIC era, patients might only have been recognized when critically ill, thus requiring hospitalization. This difference in care setting could also contribute to the readmission disparity.

Our data showed no significant difference in the time to recovery, median days hospitalized, or need for dialysis between cohorts. The similar dialysis rate (about two-thirds in both groups) likely reflects that many AKD cases were severe and occurred in patients with underlying CKD, making them severe enough to require renal replacement regardless of lab availability. These are areas where the intervention did not appear to make a measurable improvement. The similar dialysis rate (about two-thirds in both groups) likely reflects that many AKD cases were severe enough to require renal replacement regardless of lab availability. In a hospital with dialysis capability, even NIC patients would eventually receive dialysis if clinically indicated though possibly with a delay until clinical signs were obvious. The intervention didn't have a lower proportion needing dialysis compared to the NIC, which underscores that while diagnostics are crucial, they do not obviate the need for acute therapies in established AKI/AKD. However, the trend towards longer hospital stays in IC readmissions (p = 0.06) might indicate that when IC patients did get readmitted, they were managed aggressively, perhaps kept longer to thoroughly address issues. Meanwhile, NIC patients who returned might have been very acute or near end-of-life, sometimes resulting in shorter, terminal hospital stays.

One paradoxical finding was that progression to ESRD was recorded more frequently in the IC (67% vs 33% of ESRD). We interpret this not as a failure of the intervention per se, but as a reflection of two factors: survival bias and detection bias. Many NIC patients likely did not survive long enough or were not tracked long enough to document progression to ESRD. Indeed, NIC mortality was high, and those who died early would not appear as ESRD cases. In contrast, IC patients survived more and were actively followed; unfortunately, a number of them eventually progressed to ESRD

despite all efforts. This is not entirely surprising as many had underlying CKD and severe AKD insults, so some progression was inevitable. The intervention prolonged their lives and possibly slowed the decline, as suggested by the significantly longer median time to ESRD in IC ~ 5 months vs 3 weeks in NIC. But ultimately, a subset still advanced to kidney failure. In effect, the intervention succeeded in extending the window of care and forestalling death, but it "failed" to prevent the development of ESRD in many patients with intrinsically severe renal disease. This highlights an important point: while better diagnostics and supportive care improve short-term outcomes, additional strategies are needed to alter long-term disease course such as disease-specific therapies, timely referrals for transplant, etc. It is also possible that the IC's vigilant monitoring meant better detection of ESRD. Some NIC patients might have progressed to ESRD after discharge but never returned to our facility and thus were not counted. The nearly five-fold difference in recorded time to ESRD (21 days NIC vs 140 days IC) strongly indicates that NIC cases were only documented as ESRD if it happened almost immediately, whereas slower progression in NIC went unreported. Therefore, we caution against interpreting the higher ESRD rate in IC as a worse outcome as it is more accurately a consequence of improved survival and follow-up.

Examining the laboratory differences offers further insight into the cohorts. NIC patients had significantly higher WBC, monocyte, and platelet counts at presentation, suggesting more frequent systemic inflammation or infection. For instance, all pyelonephritis cases were in NIC, which could explain elevated WBC and monocytes there. High monocytes can be associated with certain infections like tuberculosis or with subacute inflammatory states, some of which were likely under-diagnosed in NIC. The IC's higher lymphocyte-to-monocyte ratio (LMR) and lower platelets align with a profile of less inflammation/stress. This could mean that the IC included relatively more patients with chronic-progressive kidney disease (e.g., hypertensive nephrosclerosis, chronic GN), whereas NIC had more acute infectious that triggers leukocytosis. It's also plausible that improved outpatient care such as treating infections early, managing volume status in the IC reduced the inflammatory burden by the time of AKD diagnosis. These differences suggest that the intervention shifted the patient case-mix, resulting in fewer acute infections and more chronic conditions being captured.

Our finding that hemoglobin level correlated with mortality (anemic patients fared worse) is consistent with other studies showing anemia is a risk factor in AKI/AKD outcomes [19, 20]. Low hemoglobin may reflect chronic kidney disease or acute hemodilution/blood loss, and it can contribute to tissue hypoxia, compounding AKI/AKD [21]. In our context, severe anemia likely signaled critical illness such as severe hemolysis, hemorrhage, or late-stage CKD. The intervention did not directly address anemia beyond diagnosis, but knowing the hemoglobin allowed appropriate treatment and management. While hemoglobin's association with mortality did not remain significant after adjustment perhaps due to the dominant effect of the intervention variable, the trend suggests that addressing anemia in AKD could be an important supportive measure. This is one aspect the intervention indirectly helped with, by providing CBC results, it enabled clinicians to recognize and treat anemia. We did not specifically track transfusions or anemia management outcomes, but future work could explore if correcting anemia improves AKD recovery.

### 4.2 Strengths and limitations

Our study has several strengths. First, by using a historical control cohort matched for basic demographics, we approximated a controlled comparison for the intervention's impact. The patients were drawn from the same hospital, with the same inclusion criteria and similar baseline characteristics, reducing confounding due to population differences. Second, our intervention was pragmatic, essentially providing routine tests and making the findings highly relevant for real-world practice and policy. The improvements in outcomes we observed could likely be replicated in similar hospitals by allocating resources for lab diagnostics. Third, we collected detailed data on a range of outcomes (both short-term and long-term) and used appropriate statistical methods given the sample size, including non-parametric tests and multivariable modeling, lending credibility to the results.

However, we acknowledge important limitations. The sample size was relatively small (n = 78), dictated by the number of AKD cases available in the time frames. This limits statistical power for some comparisons and our ability to detect

smaller differences. Despite this, the mortality difference was so large that it was statistically significant and likely clinically meaningful. Another limitation is the retrospective nature of the NIC data, which may suffer from missing information and lack of long-term follow-up. We mitigated this by excluding participants with major missing data, but it is possible that some outcomes (especially progression to ESRD) were under-ascertained in NIC. As discussed, many NIC patients may have been lost to follow-up or died at home, leading to under-reporting of events like ESRD. In contrast, the IC patients were actively followed through July 2024, so their outcomes were more completely captured. This discrepancy in follow-up could exaggerate differences in non-mortality outcomes, for example, the higher ESRD rate in IC is likely influenced by more complete follow-up. Survivorship bias is another issue, because IC patients lived longer, they inherently had more opportunity to develop ESRD or require re-hospitalization over time. We attempted to account for time-to-event through survival analysis for mortality, but for ESRD we did not perform a time-to-event analysis beyond comparing medians, so some bias may remain.

Another important limitation is that in our setting (Livingstone, Zambia), chronic dialysis resources are available but extremely limited. Not every patient who progresses to ESRD can receive long-term dialysis often due to resource and access constraints and there are no kidney transplant services. This limitation likely impacted survival: patients in either cohort who ended up with ESRD had a poor prognosis largely because only a few could continue dialysis beyond the acute period. For instance, some might get acute dialysis during hospitalization, but long-term maintenance dialysis is not guaranteed for all. The higher mortality among those with a diagnosis of ESRD is partly due to the lack of available definitive renal replacement therapy.

By matching on age and sex, we may have inadvertently introduced some selection constraints; for example, if the hospital's patient population got older over time, matching could underrepresent that trend. We argue, though, that the time gap between NIC and IC was short (~1–2 years), so large shifts in population demographics are unlikely. We did not see evidence of major socio-economic changes in that interval. We also lacked granular socio-economic data to match on (education, income, etc.). Age and sex matching were done to improve comparability, and we acknowledge that uncontrolled differences like socio-economic factors or general improvements in care over time could still exist and affect outcomes. Additionally, while we matched on age and sex, there could be residual confounding by other factors. The cohorts differed in some baseline respects such as proportion of outpatients vs inpatients, prevalence of known CKD and hypertension. It's possible that the NIC cohort patients were, on average, sicker at baseline or had more acute presentations as suggested by higher inflammatory markers and all being inpatients. If so, this could inflate the observed benefit of the intervention. However, it's worth noting that baseline kidney function (creatinine, urea, eGFR) was similarly poor in both groups, and the logistic regression adjusted for variables like initial creatinine and still found the intervention effect significant. Nonetheless, we cannot fully exclude that some of the mortality difference was influenced by unmeasured factors such as changes in overall hospital care quality over time, or differences in comorbidities that weren't recorded. Our study spanned different time periods (2021–2022 for NIC vs 2023–2024 for IC); improvements in general critical care or other treatments over those years could have contributed to better outcomes in 2023–24 independent of lab availability. We believe the intervention was the major driver given its magnitude of effect, but future studies ideally should be prospective with concurrent controls or randomized design to confirm causality.

Another limitation is that we did not formally assess implementation outcomes despite a hybrid type 3 design. We focused on clinical effectiveness; however, factors like cost of providing the lab tests, feasibility, and sustainability of this intervention are crucial for policy decisions. We reported that medication costs did not differ, but we did not measure the direct cost of lab tests supplied. In a resource-limited setting, providing a full panel of tests to every AKD patient has cost implications. Our encouraging results must be weighed against the investment required, though it's notable that many of the tests are inexpensive relative to the cost of dialysis or ICU care that might be avoided by early intervention. An economic analysis would strengthen the argument for such interventions. We also did not detail which specific tests were most impactful. It could be that a subset of the "complete profile" for example, rapid creatinine, point-of-care potassium,

and infection screening provides most of the benefit. Future work could attempt to distill the intervention to the most critical diagnostics for AKD management in low-resource hospitals.

Despite these limitations, our study clearly demonstrates that enhancing laboratory diagnostic capacity can improve patient-centered outcomes in AKD. The benefits of the intervention were multifold: fewer deaths, more accurate diagnoses, fewer misdiagnoses, and fewer hospital readmissions which collectively indicate better quality of care, Fig 4. These findings carry significant implications. In many LMIC hospitals, reagent stock-outs and limited testing are common, and clinicians often must "guess" the diagnosis or manage patients blindly. Our data provide quantitative evidence that this status quo has tangible harms including higher mortality, etc., and conversely, that investing in basic lab infrastructure or increasing laboratory test profile for patients can yield substantial improvements. In particular, acute kidney care programs in low-resource settings should prioritize reliable access to kidney function tests and related labs. This could be achieved via strengthening hospital laboratories or deploying point-of-care devices for creatinine and other key tests [17]. Training and protocols must accompany the availability of tests, for example, ensuring that doctors know how to interpret results and act swiftly. Our study inherently included an element of clinician engagement, since the presence of tests presumably prompted more active management. Additionally, community and primary care improvements are needed. The fact

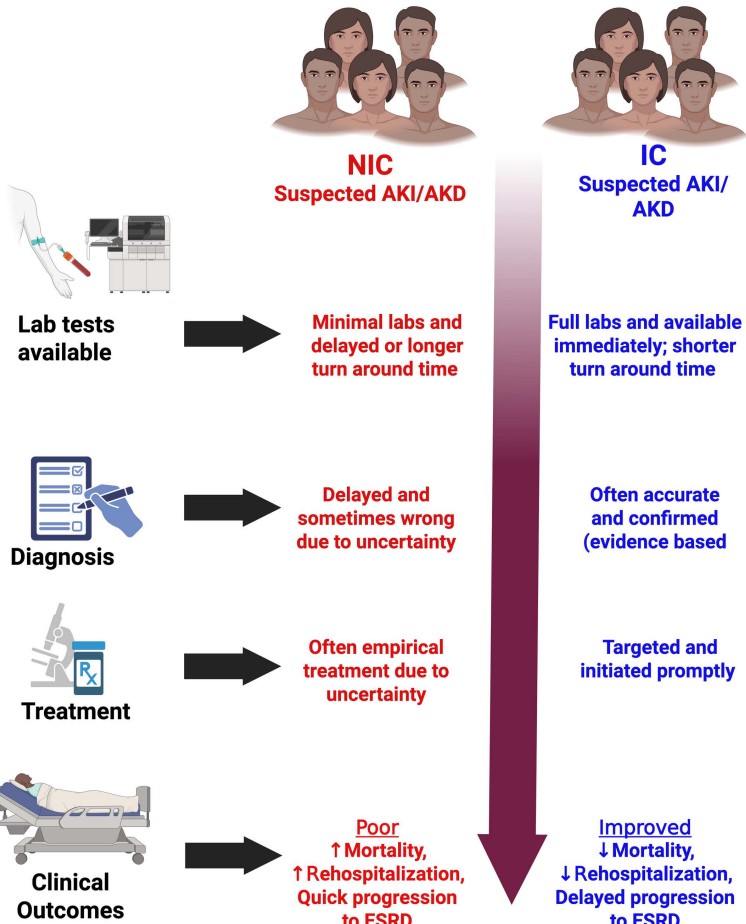

**Fig 4. Comparison of key clinical outcomes between the non-intervention (NIC) and intervention (IC) cohorts.** NIC, non-intervention cohort; IC, Intervention cohort.

that many AKD patients in NIC were only identified when severely ill indicates that earlier detection in the community was lacking. Outreach programs to educate healthcare workers about AKI/AKD warning signs, along with wider availability of simple tests (urine dipsticks, point-of-care creatinine), could help identify at-risk patients earlier and refer them before injury progresses.

Our study also underscores that while lab support can improve short-term outcomes, addressing long-term outcomes like CKD/ESRD progression remains challenging. Most of our patients had acute kidney disease superimposed on chronic damage (70% had CKD), reflecting the well-known AKI-to-CKD continuum [19]. For these patients, comprehensive care should extend beyond the acute phase. Post-AKD care clinics or follow-up systems are crucial to monitor kidney function recovery and implement measures to slow CKD progression such as renin-angiotensin system blockers for those with proteinuria, blood pressure control, etc. In the IC, we prolonged time to ESRD, which suggests that with proper follow-up, there is an opportunity to intervene in that window. A structured follow-up, perhaps facilitated by the same lab tests could improve the chances of renal recovery or at least delay ESRD further. Unfortunately, resource constraints often mean patients are lost to follow-up once the acute episode resolves. Future interventions might include not only lab tests during hospitalization but also ensuring those patients get continuity of care which can include for scheduling them in renal clinic with necessary lab work at intervals.

## 5 Conclusions

The ICLATA study demonstrated that implementing a complete clinical laboratory test profile for patients with AKD significantly improved critical outcomes, particularly survival, in a low-resource hospital. The intervention led to more accurate and timely diagnoses, reduced misdiagnoses, and better management, which translated into fewer deaths and readmissions. However, the intervention alone did not eliminate all adverse outcomes; a substantial proportion of patients still progressed to ESRD, highlighting the severity of AKD and the need for ongoing care and possibly more advanced interventions. Our findings support calls for strengthening diagnostic infrastructure as part of acute kidney care in resource-limited settings. By ensuring that no patient is treated "in the dark" due to lack of lab tests, we can make AKD not a death sentence but a survivable and manageable condition, even in low-income environments. Further research should explore the cost-effectiveness of such interventions and investigate additional strategies such as early therapeutic interventions and follow-up programs to improve long-term renal outcomes for AKD survivors. Ultimately, bridging the gap in laboratory diagnostics is a crucial step toward equity in acute kidney care and achieving the goal that no one dies of untreated acute kidney injury or disease.

## Supporting information

**S1 Checklist. STROBE checklist.**
(DOCX)

**S1 Data. Minimal underlying data required to replicate the findings reported in the manuscript.**
(XLSX)

## Author contributions

**Conceptualization:** Sepiso K. Masenga.

**Data curation:** Sepiso K. Masenga, Luyando Mutelo, Cornelius Simutanda, Lukundo Siame, Gift C. Chama, Lweendo Muchaili, Bislom C. Mweene, Situmbeko Liweleya, Sydney Mulamfu, Benson M Hamooya.

**Formal analysis:** Sepiso K. Masenga, Luyando Mutelo, Benson M Hamooya.

**Funding acquisition:** Sepiso K. Masenga.

**Investigation:** Sepiso K. Masenga, Cornelius Simutanda.

**Methodology:** Sepiso K. Masenga, Cornelius Simutanda, Lukundo Siame.

**Project administration:** Sepiso K. Masenga.

**Resources:** Sepiso K. Masenga.

**Software:** Sepiso K. Masenga.

**Supervision:** Sepiso K. Masenga.

**Validation:** Sepiso K. Masenga, Benson M Hamooya.

**Visualization:** Sepiso K. Masenga, Benson M Hamooya, Annet Kirabo.

**Writing – original draft:** Sepiso K. Masenga, Luyando Mutelo, Cornelius Simutanda, Lukundo Siame, Gift C. Chama, Lweendo Muchaili, Bislom C. Mweene, Situmbeko Liweleya, Sydney Mulamfu, Benson M Hamooya, Annet Kirabo.

**Writing – review & editing:** Sepiso K. Masenga, Luyando Mutelo, Cornelius Simutanda, Lukundo Siame, Gift C. Chama, Lweendo Muchaili, Bislom C. Mweene, Situmbeko Liweleya, Sydney Mulamfu, Benson M Hamooya, Annet Kirabo.

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
