## [Decision Letter · Decision Letter 0]

22 Apr 2025

PGPH-D-24-03008

Impact of Implementing a Complete Clinical Laboratory Test Profile on Clinical Outcomes of Acute Kidney Injury: The ICLATA Study

Dear Dr. Masenga,

Thank you for submitting your manuscript to PLOS Global Public Health. After careful consideration, we feel that it has merit but does not fully meet PLOS Global Public Health’s publication criteria as it currently stands. Therefore, we invite you to submit a revised version of the manuscript that addresses the points raised during the review process.

We look forward to receiving your revised manuscript.

Kind regards,

Valerie Ann Luyckx

Academic Editor

Journal Requirements:

1. Please include a complete copy of PLOS’ questionnaire on inclusivity in global research in your revised manuscript. Our policy for research in this area aims to improve transparency in the reporting of research performed outside of researchers’ own country or community. The policy applies to researchers who have travelled to a different country to conduct research, research with Indigenous populations or their lands, and research on cultural artefacts. The questionnaire can also be requested at the journal’s discretion for any other submissions, even if these conditions are not met.  Please find more information on the policy and a link to download a blank copy of the questionnaire here: https://journals.plos.org/globalpublichealth/s/best-practices-in-research-reporting. Please upload a completed version of your questionnaire as Supporting Information when you resubmit your manuscript. 2. Your current Financial Disclosure states, “The author(s) declare that financial support was received for the research, authorship, and/or publication of this article. This work was supported by the International Federation of Clinical Chemistry and Laboratory Medicine (IFCC)'s Task Force on Outcome Studies in Laboratory Medicine (TF-OSLM) (SKM), Fogarty International Center and National Institute of Diabetes and Digestive and Kidney Diseases of the National Institutes of Health grants R01HL144941 (AK) and 2D43TW009744 (SKM), R21TW012635 (AK and SKM) and the American Heart Association Award Number 24IVPHA1297559 https://doi.org/10.58275/AHA. 24IVPHA1297559.pc.gr.193866 (AK and SKM).”. However, your funding information on the submission form indicates that you received funding from “American Heart Association, National Institute of Diabetes and Digestive and Kidney Diseases”. Please indicate by return email the full and correct funding information for your study and confirm the order in which funding contributions should appear. Please be sure to indicate whether the funders played any role in the study design, data collection and analysis, decision to publish, or preparation of the manuscript. 3. Please provide separate figure files in .tif or .eps format. For more information about figure files please see our guidelines:  https://journals.plos.org/globalpublichealth/s/figures https://journals.plos.org/globalpublichealth/s/figures#loc-file-requirements

Additional Editor Comments (if provided):

Editor:

This study tackles an important problem which is the lack of necessary laboratory tests to diagnose kidney disease and how its lack increases the risk f poor outcomes.

In addition to the comments by the 2 reviewers there some additional concerns and issues that should be addressed:

1. How was the study funded and how much would all the tests cost

2. Were all tests done in each patient? many of these tests are not always necessary to diagnose AKI, the choice of tests should be made upon he clinical suspicion. If this was indeed the care t may need to be more explicit. It would be of interest how many and which of these tests were ordered per patient? HOw often were tests done? only on admission or throughout the hospital stay?

3. What was the duration of follow up?

4. How were the 39 NIC cases with AKI determined? It is likely these cases has laboratory testing to make these diagnoses…if yes, then what was different between these cases and the IC cases? Please explain as this is critical to the validity of the entire manuscript

5. What does a ”prompt manner” mean in terms of the intervention in the IC? How was this different from the NIC

6. Page 6 line 176 - anemia does not cause AKI, but malaria itself does

7. Many proportions are stated in the text which are repeated in the table. These do not need to be repeated in the text.

8. Statements linked in line 178: The proportion of pyelonephritis in the IC compared to the NIC significantly reduced by 100% (p < 0.0001) is confusing...rather word this as was lower...as the testing itself did not "reduced" the incidence of pyelonephritis

9. similarly, throughout the manuscript please avoid the implication that the testing "reduced" or "increased" certain diagnoses...the testing increased the chance of diagnosis and the diagnosis was lower or higher or more frequent or less frequent... these confusing statements requires significant rewriting of the manuscript. Please proof read the manuscript carefully to make these changes if necessary? These statements continue throughout the manuscript, too many to list here.

10. similarly outpateint and inpatient "cases" did not really increase or decrease...the testing may have reduce hospital admissions...how was this determined? Were things ruled out” by getting faster laboratory results?

11. please refrain from describing patients as cases

12. The statement that the testing was beneficial for outcomes may be true, but this would only be true if it led to interventions and treatments,

13. It seems there was more death in the IC cohort from Table 1? this is not a better outcome? This is opposite to what is stated in the text in several places – please recheck the numbers in the tables and the statements in the text. A Kaplan Meier curve would be of interest.

14. How was adequacy and timeliness of delivery of results measured? Line 197

15. Tabel 1 is very long, consider breaking into 2 with demographics and clinical findings in 2 tables.

16. How was Acute kidney disease defined? AKI is not in the table?

17. What does “cost of medication” in Table 1 refer to? Why is this relevant?

18. Table 2 is confusing with umping CKD, ESKD and death all together…please separate these. Is dialysis available to all who develop ESKD in your setting? IF not this must be stated as this likely contributed to deaths among those with a diagnosis

19. There is mention that much data was missing in the NIC group – please explain exactly where and when data was missing and how many were lots to follow up. These numbers may have impacted/biased the statistical conclusions?

20. Please explain what is meant by missed diagnosis and delayed diagnosis

21. How was pre-exiting CKD determined? Stated in the discussion

22. The use of nephrotoxic drugs did not differ between NIC and IC , how is this explained?

23. I disagree that pyelonephritis is likely hospital acquired, this explanation does not seem plausible

24. Overall I am still unsure of how and why the IC may have done better (if indeed they did?) – more explanation on the nature of the intervention is crucial 0 what was different between NIC and IC, how was this funded, how where results made available more promptly and what actually did this lead to in terms of change in practice? What changed at the lab, what changes in terms of transport of labs or results, payment by whom etc? This detail and ideally a flow chart of before and after would be useful to understand.

25. A major limitation is the inclusion of known AKI in NIC…this need sot be explained as well

26. eGFR is usually not used for AKI

7. The statement that the testing was beneficial for outcomes may be true, but this would only be true if it led to interventions and treatments,

8. please diving table 1 into 2 tables, demographics and results

9. It seems there was more death in the IC cohort? this is not a better outcome?

please the comments form the 2 reviewers

Reviewers' comments:

Reviewer's Responses to Questions

**Comments to the Author**

1. Does this manuscript meet PLOS Global Public Health’s publication criteria ? Is the manuscript technically sound, and do the data support the conclusions? The manuscript must describe methodologically and ethically rigorous research with conclusions that are appropriately drawn based on the data presented.

Reviewer #1: Yes

Reviewer #2: Yes

2. Has the statistical analysis been performed appropriately and rigorously?

Reviewer #1: Yes

Reviewer #2: I don't know

3. Have the authors made all data underlying the findings in their manuscript fully available (please refer to the Data Availability Statement at the start of the manuscript PDF file)?

Reviewer #1: Yes

Reviewer #2: Yes

4. Is the manuscript presented in an intelligible fashion and written in standard English?

Reviewer #1: Yes

Reviewer #2: Yes

5. Review Comments to the Author

Reviewer #1: Overall comment:

The authors intended to evaluate the impact of implementing a complete clinical laboratory test profile on clinical outcomes of Acute Kidney Injury by comparing retrospective data and the interventional data applying complete laboratory testing. The strengths of this paper is the use of a nearly complete Laboratory testing profile for early diagnosis of AKI. Although a few more accurate tests like Urinalysis mainly on electrolytes (Very important) and Cystatin C were left out, the authors tried to exhaust the testing profile. However, they did not mention the testing platforms and technologies used. What testing platforms did they use? E.g COBAS Integra 400, 800, C111, Vitros 350, Beckman Coulter and for what profiles? Otherwise, it is a good paper.

Detailed comments:

Abstract: Looks good

Introduction: Line 54 – 56: Acronyms are already defined in the abstract. Not sure it’s necessary to redefine them again.

Line 59: Urinalysis encompasses examination by microcopy, automation and RDT to determine urine osmolarity, proteins and electrolytes so I guess its repetition.

Methods: Line 82-85 Which lab tests were being considered for the NIC to determine AKI or they didn’t do any testing at all?

Line 86: Wouldn’t matching for Age and Sex intentionally reflect bias in your inclusion criteria? Maybe the Socio-economic factors between the NIC and the IC have long changed so leaving the age open I guess would still be ok in the IC.

Line 102: “The NIC lacked all the necessary laboratory support due to insufficient and lack of reagents” but in line 85, you mention there are laboratory tests that were done for the NIC. This is contradicting statement.

Line 105: “We included adults aged 18 years and above with an initial diagnosis of AKI” then what do you mean by “matching age” in Line 86?

Did you consider history of bacterial infections, surgery, use of Non-steroidal anti-inflammatory drugs for eligibility?

Line 109-110 Acronyms already defined.

Line 118: the greater than or equal to (≥) symbol is available in insert symbol.

Results: Line 161-165 Are not results. This should be in the methodology. Flowchart should also be in the methods.

Line 172 PLWH not defined anywhere in the text

Line 182-186 Misdiagnosis/ delayed diagnosis of AKI was higher in NIC but the IC was not any effective either, since it didn’t not reduce progression to outcomes. According to your results, outcomes were more in the IC (maybe because they were detectable). I believe you should bring this point out well in the discussion.

Line 187-190 Rehospitalization is not mentioned anywhere as an outcome/ dependent variable. You only have CKD, ESRD and Death.

Line 241 How does absence of renal infection be associated with adverse outcomes. It doesn’t make sense. In my view, just don’t mention it. You do not have to mention everything from your analysis. Some associations between variables don’t make sense sometimes.

Table 1 and Table 2: How do we distinguish these 2 tables? Clinical characteristics and Clinical factors. Wouldn’t association before and after intervention be sufficient? Find a way of merging these 2 tables if they are not any different.

Also find a way of merging Table 3 and Table 4. I think they are communicating the same info.

In your discussion and conclusion, include that fact that progression to the outcomes despite early diagnosis, was not deterred or reduced in the IC

Line 312-313 What do you mean improves morbidity and mortality? Or you meant reduces morbidity and mortality?

Fig. 1 Is too abstract and is incomplete.

Create some graphical representations of association of clinical factors with adverse outcomes to easily comprehend information given in the tables. This is just a suggestion otherwise it’s a very informative manuscript.

Reviewer #2: The study, Impact of implementing a complete clinical laboratory test profile on clinical outcomes of acute kidney injury: The ICLATA study, was an interventional one. The study design blended clinical effectiveness with implementation science to determine the impact of implementing a complete clinical laboratory test profile on AKI clinical outcomes using a mixed methods model, comparing retrospective and prospective AKI cohorts. This gave a richer, more comprehensive understanding of the impact in a low-resource setting where comprehensive laboratory testing is usually unavailable. The findings reported in the paper are quite relevant in the context of an issue as topical as acute kidney injury. The authors found that the provision of immediate, more comprehensive laboratory tests (in the intervention cohort) was associated with increased odds of identifying the cause of AKI, increased CKD diagnosis, reduced AKI misdiagnosis or delayed diagnosis, as well as reduced mortality.

The manuscript is well-organized, written clearly, and follows the journal's publishing criteria. It is my impression that the manuscript is technically sound and reproducible and conforms to the STROBE guidelines. However, a larger sample size, especially on the quantitative side, would have more robustly evaluated the process and built a stronger case for scaling up. The limitation of the small sample size has already been alluded to by the authors. The study can be replicated.

It appears that the statistics is appropriately done. However, I am not well-versed in all aspects of statistics with respect to this study design and thus my expertise in assessing this section is limited.

It is unclear how the authors differentiated between 'renal infection' and 'pyelonephritis'.

The results are well presented and easy to understand.

There was no robust discussion on costs.

The manuscript is generally well written.

6. PLOS authors have the option to publish the peer review history of their article (what does this mean? ). If published, this will include your full peer review and any attached files.

**Do you want your identity to be public for this peer review?** For information about this choice, including consent withdrawal, please see our Privacy Policy .

Reviewer #1: **Yes: ** Brian A. Kagurusi

Reviewer #2: No

---

## [Decision Letter · Decision Letter 1]

1 Sep 2025

PGPH-D-24-03008R1

Reducing Mortality in Acute Kidney Disease through Comprehensive Laboratory Support: Findings from the ICLATA Hybrid Implementation Study in Zambia

Dear Dr. Masenga,

Thank you for submitting your manuscript to PLOS Global Public Health. After careful consideration, we feel that it has merit but does not fully meet PLOS Global Public Health’s publication criteria as it currently stands. Therefore, we invite you to submit a revised version of the manuscript that addresses the points raised during the review process.

The reviewer has left some further feedback for your consideration. Please provide a point by point response to their comments.

We look forward to receiving your revised manuscript.

Kind regards,

Joanna Tindall, PhD

Staff Editor

Journal Requirements:

Additional Editor Comments (if provided):

Reviewers' comments:

Reviewer's Responses to Questions

**Comments to the Author**

1. If the authors have adequately addressed your comments raised in a previous round of review and you feel that this manuscript is now acceptable for publication, you may indicate that here to bypass the “Comments to the Author” section, enter your conflict of interest statement in the “Confidential to Editor” section, and submit your "Accept" recommendation.

Reviewer #2: All comments have been addressed

2. Does this manuscript meet PLOS Global Public Health’s publication criteria ? Is the manuscript technically sound, and do the data support the conclusions? The manuscript must describe methodologically and ethically rigorous research with conclusions that are appropriately drawn based on the data presented.

Reviewer #2: Yes

3. Has the statistical analysis been performed appropriately and rigorously?

Reviewer #2: Yes

4. Have the authors made all data underlying the findings in their manuscript fully available (please refer to the Data Availability Statement at the start of the manuscript PDF file)?

Reviewer #2: Yes

5. Is the manuscript presented in an intelligible fashion and written in standard English?

Reviewer #2: Yes

6. Review Comments to the Author

Reviewer #2: This study addresses an important problem in low-resource settings: diagnostic capacity as it concerns acute kidney disease. The manuscript is clearly written and fairly easy to understand. It represents a clear improvement over the previous submission.

The methodology section is well described, more scientifically-sounding and the flow is much improved. The statistical methods used are appropriate.

The representation of the results is much improved and clearer to the reader.

Table 1 should be retitled appropriately. The way it is written currently sounds as if it is the same set of study participants who were tested before an intervention, and again after the intervention. This is not the case.

In the row referring to suspected underlying cause of AKD, all the identifiable causes should be listed first. The ‘Other’ should be placed in the last row, below the others.

Table 2 is wrongly titled. These are laboratory parameters, and not sociodemographic or clinical factors.

Figure 4 can be better titled. It appears to be a comparison and the figure title should reflect that.

The discussion and conclusion sections are well balanced and adequately supported by the data. It was stated (page 17, line 464) that many of the IC patients had underlying CKD. The determination of the ‘underlying CKD’ is unclear.

There are quite a number of limitations. However, they are clearly stated.

The abstract accurately conveys what has been found.

7. PLOS authors have the option to publish the peer review history of their article (what does this mean? ). If published, this will include your full peer review and any attached files.

**Do you want your identity to be public for this peer review?** For information about this choice, including consent withdrawal, please see our Privacy Policy .

Reviewer #2: No

---

## [Editor Report · Decision Letter 2]

3 Oct 2025

Reducing Mortality in Acute Kidney Disease through Comprehensive Laboratory Support: Findings from the ICLATA Hybrid Implementation Study in Zambia

PGPH-D-24-03008R2

Dear Prof. Masenga,

We are pleased to inform you that your manuscript 'Reducing Mortality in Acute Kidney Disease through Comprehensive Laboratory Support: Findings from the ICLATA Hybrid Implementation Study in Zambia' has been provisionally accepted for publication in PLOS Global Public Health.

Best regards,

Julia Robinson

Executive Editor